# Structural determinants of nuclear export signal orientation in binding to exportin CRM1

Ho Yee Joyce Fung[1], Szu-Chin Fu[1], Chad A Brautigam[2], Yuh Min Chook[1]*

[1]Department of Pharmacology, University of Texas Southwestern Medical Center, Dallas, United States; [2]Department of Biophysics, University of Texas Southwestern Medical Center, Dallas, United States

**Abstract** The Chromosome Region of Maintenance 1 (CRM1) protein mediates nuclear export of hundreds of proteins through recognition of their nuclear export signals (NESs), which are highly variable in sequence and structure. The plasticity of the CRM1-NES interaction is not well understood, as there are many NES sequences that seem incompatible with structures of the NES-bound CRM1 groove. Crystal structures of CRM1 bound to two different NESs with unusual sequences showed the NES peptides binding the CRM1 groove in the opposite orientation (minus) to that of previously studied NESs (plus). Comparison of minus and plus NESs identified structural and sequence determinants for NES orientation. The binding of NESs to CRM1 in both orientations results in a large expansion in NES consensus patterns and therefore a corresponding expansion of potential NESs in the proteome.

## Introduction

The exportin CRM1 (Chromosome Region Maintenance 1 protein; also known as exportin 1 or XPO1) is the most prominent nuclear export receptor in the cell. CRM1 maintains the cellular localization of hundreds of diverse-functioning protein cargos, including many tumor suppressor, cell cycle proteins, and viral proteins (*Fornerod et al., 1997*; *Fukuda et al., 1997*; *Ossareh-Nazari et al., 1997*). CRM1 is also a promising cancer drug target, and a small molecule inhibitor of CRM1 named Selinexor is currently in more than 40 clinical trials for a variety of cancers (clinicaltrials.gov) (*Lapalombella et al., 2012*; *Etchin et al., 2013*; *Sun et al., 2013*; *Fung and Chook, 2014*; *Xu et al., 2015*). CRM1 recognizes its protein cargos through 8–15 residue long nuclear export signals (NESs) in the proteins (*la Cour et al., 2004*; *Kosugi et al., 2008*; *Xu et al., 2010*). NES sequences are highly diverse, and the peptides bind CRM1 with a large affinity range, with dissociation constants ($K_Ds$) ranging from low nanomolar to tens of micromolar (*Kutay and Guttinger, 2005*). Sequence, peptide-library, and bioinformatic analyses have found that NESs are best described by a set of six consensus sequences, which differ in the spacings between four key hydrophobic residues $\Phi1$, $\Phi2$, $\Phi3$, and $\Phi4$ (*Figure 1A*) (*la Cour et al., 2004*; *Kosugi et al., 2008*; *Xu, Farmer et al., 2012a*). While sequence patterns are available to describe many NESs, there is limited structural information on diverse NESs and how they bind CRM1.

Structures are available for only three different NESs, from the cargos protein kinase A inhibitor (PKI), Snurportin-1 (SNUPN), and the HIV1-Rev protein, bound to CRM1. The NESs bind in a hydrophobic groove, which is located on the outer/convex surface of the ring-shaped CRM1 (*Dong et al., 2009*; *Monecke et al., 2009*; *Güttler et al., 2010*). The NESs use almost exclusively their side chains, especially their hydrophobic $\Phi$ side chains, to bind CRM1. The NES-binding groove of CRM1, which is wide at one end and narrow at the other end, consists of 5 hydrophobic pockets P0–P4 and is

*For correspondence: yuhmin. chook@utsouthwestern.edu

Competing interests: The authors declare that no competing interests exist.

**eLife digest** Many organisms keep their DNA within a structure inside their cells called the nucleus. Two layers of membrane surround the nucleus and keep the DNA separate from the rest of the cell's contents. Yet, proteins and other molecules can move in and out of the nucleus by passing through small pores in this nuclear membrane.

To travel through these pores, larger molecules such as proteins rely on the assistance of transport receptors, including one called CRM1. This transport receptor helps to export hundreds of different proteins from the nucleus by recognizing a part of their structure called the 'nuclear export signal'. Earlier work has shown that three different nuclear export signals interact with CRM1 in a similar ways by binding to a groove on its outer surface. But, there are several different types of nuclear export signal, and many are predicted to have three-dimensional structures that would seem to prevent them from binding to CRM1 in this way. As yet, it remains unknown how these diverse signals interact with this important transporter receptor.

Protein crystallization is a technique that is used to visualize a protein's three-dimensional structure. Fung et al. have now used this approach to investigate how a particular class of nuclear export signals (called 'class 3') bind to CRM1. First, a modified form of CRM1 was crystallized once it had bound to a small fragment of protein that contains a class 3 nuclear export signal. The protein's molecular structure was then revealed by performing X-ray diffraction on the crystals.

The results show, unexpectedly, that two different nuclear export signals in class 3 bind to the groove of CRM1 in the opposite direction to that reported previously. Additional biochemical and structural experiments then identified a particular feature or motif in the nuclear export signals that determines which way round they bind to CRM1.

This discovery advances our understanding of how these signals work, which will allow us to more accurately identify new nuclear export signals from genome sequences. As more CRM1-binding nuclear export signals are discovered in the future, the experimental data sets used to train the computational programs that are currently used to locate these signals in genomic sequences will be diversified and improved.

virtually identical in all CRM1-NES structures (*Figure 1B*). NESs from PKI and SNUPN (PKI[NES] and SNUPN[NES]) share a similar structure when bound to CRM1—an N-terminal 3-turn α-helix followed by a short C-terminal β-strand-like extension (*Figure 1B*) (*Dong et al., 2009*; *Monecke et al., 2009*; *Güttler et al., 2010*; *Koyama et al., 2014*). The NES helix binds the wide part of the CRM1 groove, while the β-strand binds the narrow end of the groove. The Rev[NES] peptide binds the CRM1 groove in a different manner by adopting an entirely extended conformation (*Güttler et al., 2010*). All three NES peptides bind in the same direction, with their N-termini at the wide part of the groove.

The vastly different conformations of the extended Rev[NES] compared to the helix-strand PKI[NES] and SNUPN[NES] suggest that CRM1 may recognize divergent signal sequences in part by binding different peptide structures. The repertoire of conformations for CRM1-bound NESs remains unclear, but the asymmetric and seemingly structurally invariant NES-bound CRM1 groove presents physical constraints on structures of bound NESs. For example, the class 3 NES consensus of $\Phi 1X_{(2,3)}\Phi 2X_{(2,3)}\Phi 3X_2\Phi 4$ with two intervening residues between $\Phi 3$ and $\Phi 4$ suggests a single long NES helix. The substitution of a narrow strand or extended chain at the C-terminus of an NES with a helix presents a steric problem as the thicker helix is unlikely to fit into the tapering CRM1 groove. In current NES databases, class 3 NES sequences are as prevalent as NESs of classes 1b, 1c, 1d, and 2, but information of how they are able to bind CRM1 is missing (*Xu et al., 2015*).

We have developed a general strategy to crystallize CRM1 bound to NES peptides in order to study how diverse sequences, including the enigmatic class 3 NESs, bind the exportin. Crystal structures of two different class 3 NESs bound to CRM1 revealed a novel NES binding mode where polypeptide direction of the NES is reversed. We show that NES peptides can bind the CRM1 groove bidirectionally (in both plus and minus directions), and biochemical and structural analyses identified determinants for one direction of binding vs the other. Bidirectional exportin-signal interactions suggest a significant expansion of the current NES consensus patterns that will enable new, previously unknown NESs to be identified.

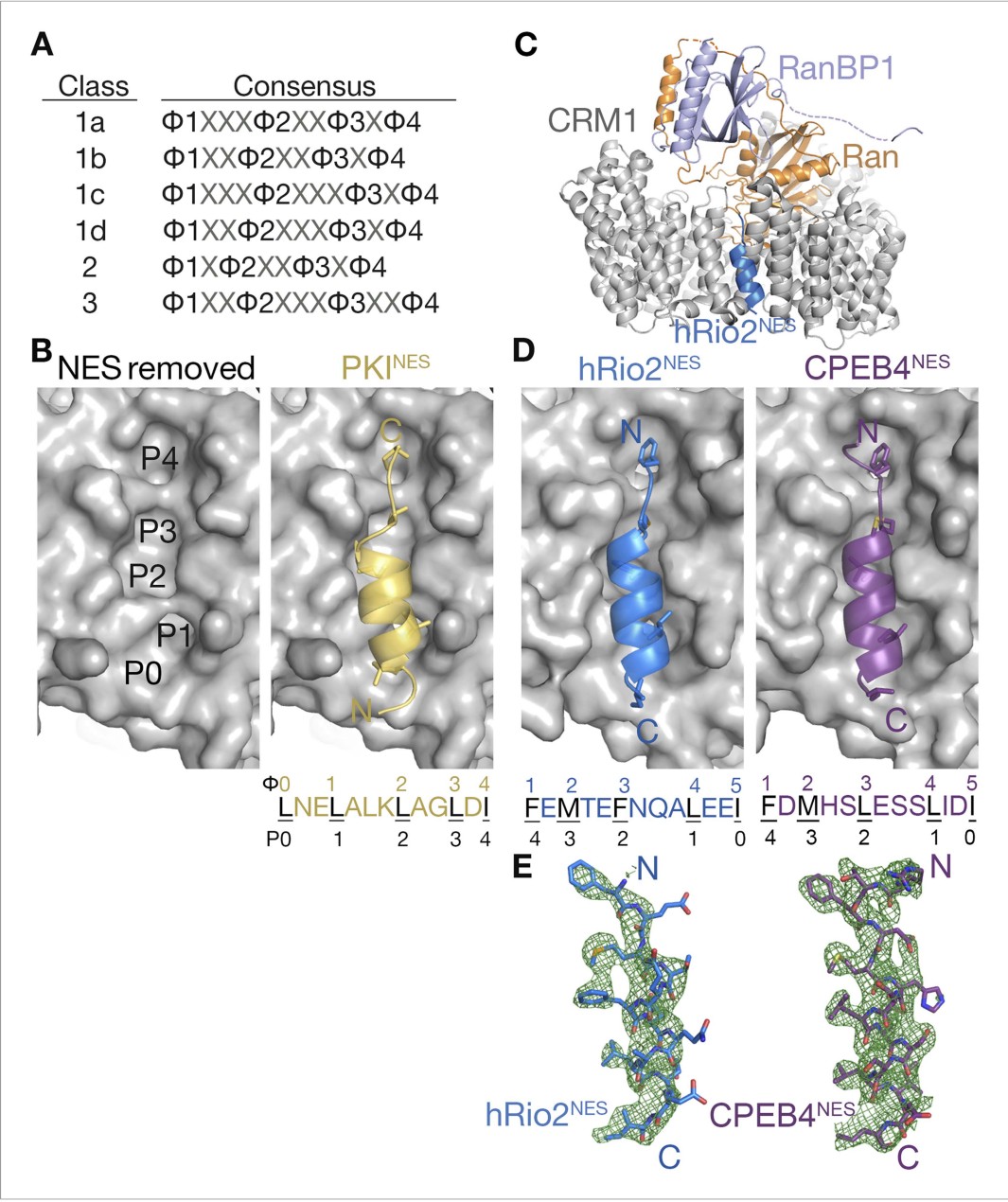

**Figure 1**. hRio2[NES] and CPEB4[NES] bind CRM1 in orientation opposite to the PKI[NES]. (**A**) Six nuclear export signal (NES) consensus patterns (Φ is Leu, Val, Ile, Phe or Met; X is any amino acid). (**B**) Structure of PKI[NES] (yellow cartoon) bound to Chromosome Region of Maintenance (CRM1) (gray surface) (3NBY) on the right and PKI[NES] was removed to show hydrophobic pockets P0–P4 in the CRM1 groove on the left panel. (**C**) Overall structure of the CRM1* (gray)-RanGppNHp (orange)-RanBP1 (light purple)-hRio2[NES] (blue) complex. (**D**) Structures of hRio2[NES] (blue) and CPEB4[NES] (purple) bound to the CRM1 groove (gray surfaces). All NES peptides are in cartoon and their hydrophobic Φ residues shown as sticks. Their Φ residues and the corresponding P0–P4 CRM1 pockets that they bind are shown below. (**E**) Kick OMIT map meshes contoured at the 3.0σ level overlaid on the final, refined coordinates for hRio2[NES] and CPEB4[NES]. Kicked OMIT maps were generated by PHENIX by omitting the NES peptides.

The following figure supplement is available for figure 1:

**Figure supplement 1**. Electron densities of the wild-type NES peptides.

## Results

### A general strategy for structure determination of CRM1 bound to NES peptides

Crystallization of CRM1-Ran-NES peptide complexes has generally not been successful, possibly due to conformational flexibility and low affinities for the NESs (Kutay and Guttinger, 2005). Crystal structures of NESs bound to CRM1 were instead determined using the CRM1-Ran-SNUPN complexes, including ones where the SNUPN[NES] was replaced with the PKI[NES] and Rev[NES] (Güttler et al., 2010). This strategy was limited because of severe mosaicity of the CRM1-Ran-SNUPN crystals (Güttler et al., 2010). On the other hand, the ternary complex of Saccharomyces cerevisiae CRM1 ([Sc]CRM1) with RanGTP (human or yeast RanGTP, Gsp1p) and RanBP1 (Yrb1p) reliably yields crystals that diffract to high resolution and has been used to determine structures of several CRM1-inhibitor complexes (Koyama and Matsuura, 2010; Lapalombella et al., 2012; Etchin et al., 2013; Sun et al., 2013; Haines et al., 2015). We therefore used the CRM1-Ran-RanBP1 complex to determine structures of the exportin bound to the enigmatic class 3 NES peptides.

RanBP1 binding normally stimulates NES release by closing the NES-binding groove of [Sc]CRM1 (Koyama and Matsuura, 2010). We engineered CRM1 to shift the open-closed groove equilibrium toward the open state, in order for CRM1-Ran-RanBP1 to bind NESs. We started with a [Sc]CRM1 construct (residues 1–1058, Δ377–413, [537]DLTVK[541] to GLCEQ) that is known to crystallize easily and has an NES groove that is virtually identical to that of human CRM1 (Lapalombella et al., 2012; Etchin et al., 2013; Sun et al., 2013; Haines et al., 2015). Koyama and Matsuura showed that mutation of the H9 loop of CRM1, which packs against the back of a closed NES groove, stabilizes the open CRM1 groove (Koyama and Matsuura, 2010). Thus, we mutated the H9 loop (Val441Asp) to detach it from the back of the NES groove and to open the groove even when CRM1 is complexed with Ran and RanBP1. The resulting CRM1* construct, with [Sc]CRM1 residues 1–1058, Δ377–413, V441D and groove residues [537]DLTVK[541] mutated to GLCEQ (to mimic human CRM1, see methods), binds NES peptides in the presence of RanGTP and RanBP1. We generated quaternary CRM1*-RanGppNHp-RanBP1-NES complexes with two class 3 NES peptides, the hRio2[NES] ([389]RSFE<u>M</u>TE<u>F</u>NQA<u>L</u>EE<u>I</u>[403]) and the CPEB4[NES] ([379]RTFD<u>M</u>HS<u>L</u>ESS<u>L</u>ID<u>I</u>[393]; predicted Φ1–Φ4 positions are underlined) and determined their structures to 2.3 Å and 2.1 Å resolution (Figure 1C,D; crystallographic statistics in Table 1). The crystals are isomorphous to previously crystallized inhibitor-bound CRM1-Ran-RanBP1 complexes, with one CRM1*-RanGppNHp-RanBP1-NES complex in the asymmetric unit (Sun et al., 2013). Residues modeled in the three proteins are CRM1* residues 1–440 and 460–1053, Ran residues 9–216, and RanBP1 residues 63–69 and 78–200. hRio2[NES] residues 391–403 and CPEB4[NES] residues 379–393 were modeled in the respective structures.

Overall structures of the CRM1*-Ran-RanBP1-NES complexes are highly similar to previously determined CRM1-Ran-RanBP1 structures (all residue Cα rmsds 0.2–0.5 Å when compared to unliganded CRM1-Ran-RanBP1 (PDB code: 3M1I, 4HB2) (Koyama and Matsuura, 2010; Sun et al., 2013) and to inhibitor-bound CRM1-Ran-RanBP1 complexes (PDB code: 4HAT, 4HAU, 4HAV, 4GMX, 4GPT) (Lapalombella et al., 2012; Etchin et al., 2013; Sun et al., 2013; Haines et al., 2015). Structures of the NES peptides were verified by kick-OMIT maps (Praznikar et al., 2009) generated without the peptide (Figure 1E, stereo views in Figure 1—figure supplement 1). Selenomethionine hRio2[NES] peptide was also generated and anomalous data were collected to confirm correct placement of its methionine, and unambiguously confirm the direction of the NES polypeptide chain (Figure 1—figure supplement 1).

### hRio2[NES] and CPEB4[NES] bind CRM1 in the opposite or minus direction

The CRM1-bound hRio2[NES] and CPEB4[NES] structures unexpectedly revealed that both NESs bound the CRM1 groove in opposite orientation (termed the minus direction) compared to previous NES structures (PKI[NES], SNUPN[NES], and Rev[NES] bind in the plus direction) (Figure 1B,D). The NES groove of CRM1 is nearly invariant when bound to plus or minus NESs (Cα rmsds 0.3–0.5 Å; all atom rmsds 1.0–1.3 Å, for CRM1 residues 521–605 in all available CRM1-NES structures [Dong et al., 2009; Monecke et al., 2009; Güttler et al., 2010]). Although the polypeptide directions are reversed, local structures of the hRio2[NES] and CPEB4[NES] are similar to those of the PKI[NES] and SNUPN[NES]. All four NES peptides are combinations of 3-turn α-helices and 2-residue β-strand-like extensions.

**Table 1.** Data collection and refinement statistics

| | ScXPO1-RanGppNHp-Yrb1p bound to NES of: | | | | |
| | Selenomethione-hRio2 | CPEB4 | Selenomethione-hRio2-R | CPEB4-R | PKI-Flip3 |
|---|---|---|---|---|---|
| Data collection | | | | | |
| Space group | P4$_3$2$_1$2 | | | | |
| Cell dimensions | | | | | |
| a, b, c (Å) | 106.48, 106.48, 303.73 | 105.96, 105.96, 304.00 | 106.69, 106.69, 304.50 | 106.48, 106.48, 303.73 | 105.96, 105.96, 304.00 |
| a, b, g (°) | 90, 90, 90 | 90, 90, 90 | 90, 90, 90 | 90, 90, 90 | 90, 90, 90 |
| Resolution (Å) | 50.00–2.28 (2.32–2.28)* | 50.00–2.10 (2.14–2.10) | 50.00–2.28 (2.32–2.28) | 50.00–2.94 (3.00–2.94) | 50.00–2.55 (2.59–2.55) |
| $R_{pim}$ | 2.9 (37.7) | 3.5 (43.4) | 3.5 (38.6) | 4.9 (40.6) | 4.1 (46.5) |
| I/sI | 24.3 (2.17) | 19.5 (1.70) | 22.5 (2.72) | 13.3 (1.87) | 19.0 (1.92) |
| Completeness (%) | 98.6 (99.8) | 99.5(100) | 98.0 (99.2) | 94.6 (96.0) | 99.6 (100) |
| Redundancy | 7.0 (5.9) | 6.0 (6.1) | 7.0 (7.0) | 6.2 (5.7) | 5.5 (5.5) |
| Refinement | | | | | |
| Resolution (Å) | 45.7–2.28 (2.32–2.28) | 40.2–2.09 (2.12–2.09) | 37.7–2.28 (2.31–2.28) | 47.5–2.94 (3.02–2.94) | 47.5–2.54 (2.60–2.54) |
| No. reflections | 77,245 (2833) | 98,659 (1793) | 79,492 (3267) | 34,265 (2013) | 56862 (3361) |
| $R_{work}$/$R_{free}$ | 17.8 (25.8)/21.9 (27.3) | 17.0 (23.8)/20.8 (27.0) | 16.8 (24.7)/21.2 (27.6) | 18.1 (25.2)/24.0 (31.3) | 18.6 (25.0)/22.6 (30.6) |
| No. atoms | | | | | |
| Protein | 10,859 | 11,114 | 10,823 | 10,708 | 10797 |
| Ligand/ion | 60 | 76 | 59 | 51 | 51 |
| Water | 271 | 660 | 358 | 8 | 253 |
| NES Peptide/Φ | 111/46 | 122/43 | 130/46 | 112/43 | 105/43 |
| B-factors | | | | | |
| Protein | 42.0 | 39.3 | 43.9 | 53.8 | 46.5 |
| Ligand/ion | 44.3 | 51.7 | 46.9 | 41.8 | 41.6 |
| Water | 33.4 | 34.8 | 35.4 | 23.3 | 35.3 |
| NES peptide/Φ | 80.5/77.3 | 77.6/70.4 | 67.5/61.7 | 81.2/80.5 | 98.6/96.0 |
| R.m.s deviations | | | | | |
| Bond lengths (Å) | 0.003 | 0.003 | 0.006 | 0.003 | 0.004 |
| Bond angles (°) | 0.617 | 0.689 | 0.835 | 0.578 | 0.673 |
| PDB code | 5DHF | 5DIF | 5DI9 | 5DHA | 5DH9 |

*Highest resolution shell is shown in parenthesis.

One crystal was used for each structure.

Helices of the minus NESs are now at the C-termini of the peptides and their strands at the N-termini (*Figure 1D*). Both plus and minus NES helices bind the same part of the CRM1 groove, with hydrophobic residues from one helix face occupying hydrophobic pockets P0–P3 of CRM1 (*Figure 1B,D*). The structures show that the two minus NESs clearly match the consensus pattern Φ1XΦ2XXXΦ3XXΦ4XXΦ5, which is the reverse of the class 1a pattern, Φ0XXΦ1XXΦ2XXXΦ3XΦ4. The five hydrophobic residues of the hRio2[NES] and CPEB4[NES], designated Φ1–Φ5, bind the same P0–P4 pockets as the plus NESs, but in reverse order, with Φ1 in P4 and Φ5 in P0 (*Figure 1B,D*). The narrow part of the CRM1 groove is still occupied by an extended strand motif, which is formed by Φ1 and Φ2 of the minus NESs as they occupy the CRM1 P4 and P3 pockets, respectively. The hRio2[NES] and CPEB4[NES] are in fact not class 3 NESs in the traditional sense, as the four previously designated Φ residues in hRio2[NES] ([389]RSFE*MTEF*NQA*LEEI*[403]) and the CPEB4[NES] ([379]RTFD*MHSLESSLIDI*[393]; predicted Φ1–Φ4 positions are underlined) do not occupy the P1–P4 hydrophobic pockets as predicted. The four hydrophobic residues that match the class 3 NES consensus, in fact form only a portion of an inverted class 1a pattern Φ2XXXΦ3XXΦ4XXΦ5, with M393 of hRio2[NES] and M383 of CPEB4[NES] in the

Φ2 positions. F391 of hRio2$^{NES}$ and F381 of CPEB4$^{NES}$, which we had previously missed as consensus residues, are the Φ1 positions of the N-terminal ΦXΦ motif of an inverted class 1a pattern.

## Comparative structural and biochemical analyses of minus and plus NESs

Comparison of plus and minus NESs showed translational offsets of the helices along their axes (*Figure 2*). Cαs of minus NESs are shifted 1.3–3.5 Å from equivalent Cαs in the plus NESs, with the largest shifts observed for residues that occupy the P0 and P1 CRM1 pockets. In an α-helix, amino acid side chains are angled toward the N-terminus of the helix. Thus, since plus and minus helices progress from N- to C-terminus in opposite directions, side chains that emanate from the helices also project in opposite directions. The Φ0, Φ1, Φ2, and Φ3 side chains of the plus NES helix project toward the wide end of the CRM1 groove near P0 to occupy the P0–P4 CRM1 pockets (*Figure 2*). In contrast, the equivalent Φ5, Φ4, Φ3, and Φ2 residues of the minus NES helix project toward the narrow end of the CRM1 groove, thus necessitating a shift of the entire helix in the opposite direction to allow the hydrophobic side chains to reach the P0, P1, P2, and P3 CRM1 pockets.

Because the entire minus NES helix shifts relative to a plus helix, it is not surprising that hydrophobic side chain preferences of the minus and plus helices are similar. We generated single amino acid mutants by replacing each of positions Φ3, Φ4, and Φ5 in hRio2$^{NES}$ with other hydrophobic residues, and tested binding of the mutants to CRM1. Results of in vitro pull-down assays using immobilized GST-hRio2$^{NES}$ mutants, purified human CRM1, and yeast RanGTP show that medium-sized hydrophobic side chains such as isoleucine and leucine are preferred at Φ4 and Φ5 for CRM1 interaction. Medium and larger hydrophobic side chains such as isoleucine, leucine, and methionine are preferred at Φ3 for binding CRM1 (*Figure 3A*). These results are similar to ones previously shown in the mutagenesis study of the PKI$^{NES}$ (*Güttler et al., 2010*).

The shift of the minus NES helix relative to the plus NES helix results in a corresponding translation of the preceding strand/loop segment that places the Φ1 and Φ2 side chains farther from the P3 and P4 pockets. In both the hRio2$^{NES}$ and CPEB4$^{NES}$, large hydrophobic residues in the Φ1 (phenylalanines) and the Φ2 (methionines) positions within the extended segments enable a longer reach into the comparatively distal P3 and P4 CRM1 pockets. Mutagenesis of the Φ1 and Φ2 positions of hRio2$^{NES}$ and pull-down assays with CRM1 and Ran show that large hydrophobic side chains such as leucine, methionine, phenylalanine, and tryptophan are preferred in both positions (*Figure 3B*). Smaller hydrophobic side chains like alanine, valine, and isoleucine in these positions are disfavored as the mutants do not bind CRM1 efficiently (*Figure 3B*). The preference for large hydrophobic side chains is consistent with the need for side chains in the extended portions of the minus NESs to reach farther into their CRM1-binding sites. In contrast, the large aromatic residues phenylalanine and tryptophan are disfavored in equivalent Φ3 and Φ4 positions in the plus direction PKI$^{NES}$ (*Güttler et al., 2010*).

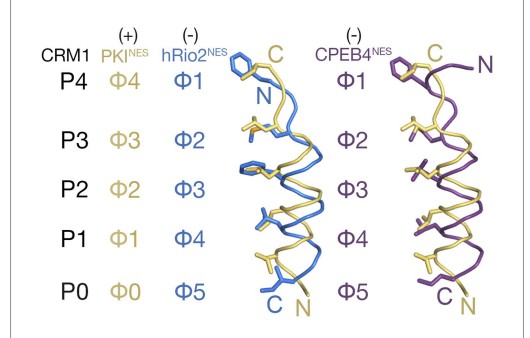

**Figure 2**. Comparison of plus and minus NESs. Pairwise comparison of hRio2$^{NES}$ (blue) or CPEB4$^{NES}$ (purple) with PKI$^{NES}$ (yellow; 3NBY) upon superposition of NES-bound CRM1 grooves. Hydrophobic NES residues (Φs) are shown as sticks and orientation of the CRM1 grooves is indicated by positions of the P0–P4 pockets.

## Structural determinants of the plus vs minus NES

Structural and mutagenesis data to compare plus and minus NESs suggest that placement of the strand-like ΦXΦ motif at the N-terminus of an NES generates a signal peptide that binds CRM1 in the minus direction, whereas a C-terminal ΦXΦ results in a plus direction NES. To first investigate whether features of the sequence such as spacings between hydrophobic residues and placement of the ΦXΦ motif are critical in determining directionality of NES binding, we reversed the sequence of the hRio2$^{NES}$ (FEMTEFNQALEEI) to generate hRio2$^{NES}$-R (IEELAQNFETMEF). We also reversed CPEB4$^{NES}$ (FDMHSLESSLIDI) to generate CPEB4$^{NES}$-R (IDILSSELSHMDF). Both reversed peptides match the class 1a NES pattern and were predicted to bind CRM1 like the PKI$^{NES}$, which is another class 1a NES. Binding affinities of NES peptides to

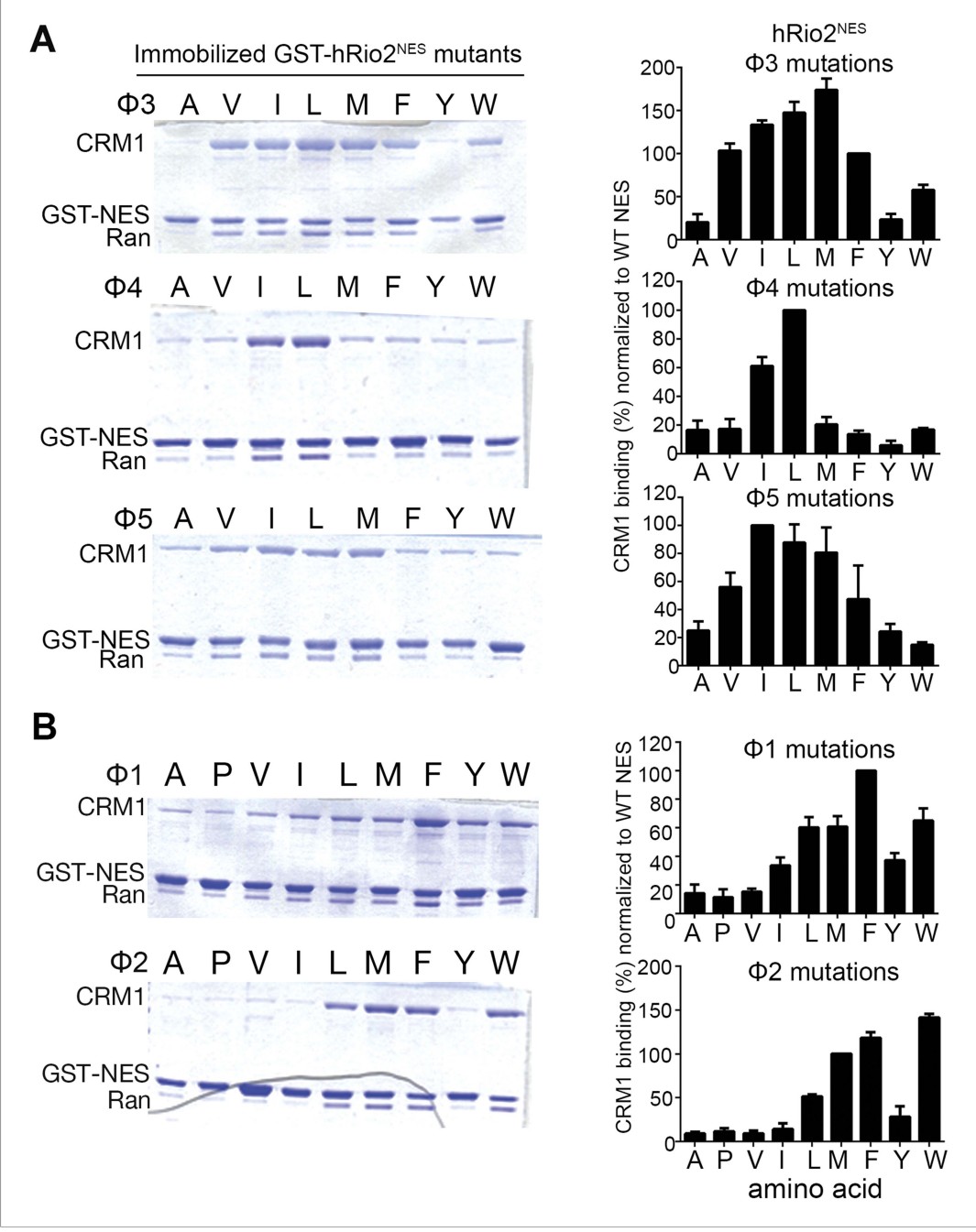

**Figure 3.** Hydrophobic side chain preferences for hRio2[NES] binding to CRM1. In vitro pull-down assay (Coomassie-stained SDS/PAGE) of purified human CRM1 binding to immobilized GST-hRio2[NES] mutants (**A**) Φ3, Φ4, or Φ5 or (**B**) Φ1 or Φ2 position mutated in the presence of excess [Sc]RanGTP. Relative band intensities of triplicate experiments are plotted in histograms.

CRM1 were measured in competition differential bleaching experiments using FITC-PKI[NES] as a fluorescent probe, MBP-NESs as competitors and monitored with a microscale thermophoresis instrument (*Figure 4*, *Figure 4—figure supplement 1*). The competition differential bleaching approach is explained in methods and representative titration data are shown in *Figure 4—figure supplement 2*. Wild-type NESs MBP-hRio2[NES] and MBP-CPEB4[NES] bind CRM1 with $K_D$s of 2200 nM [1600,2900] and 590 nM [400,840], respectively (*Figure 4B*). The ranges in brackets represent the 68.3% confidence intervals

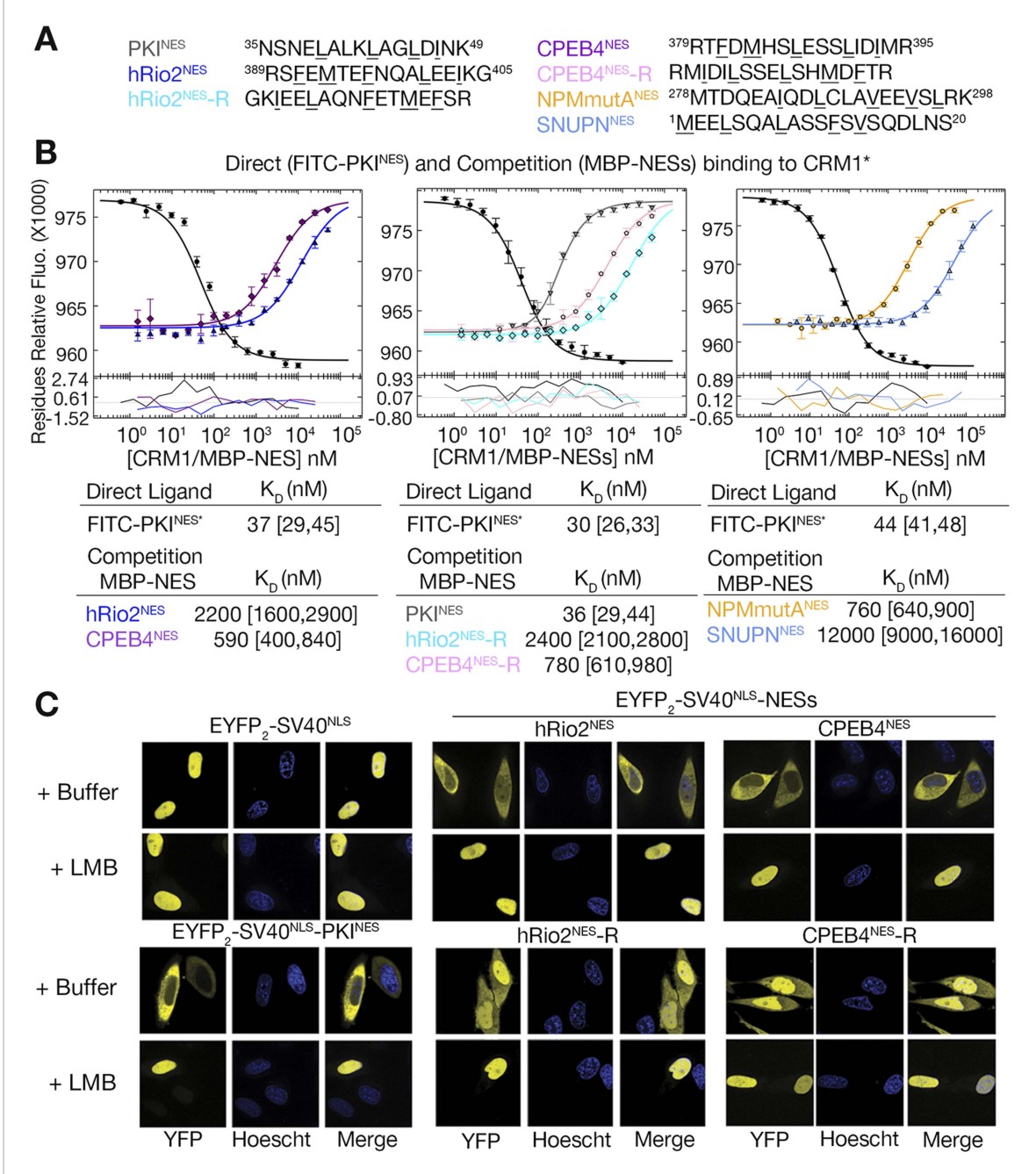

**Figure 4**. hRio2[NES], CPEB4[NES], and their reverse counterparts bind CRM1 with similar binding affinities. (**A**) Sequences of NESs used. (**B**) Binding of FITC-PKI[NES] and various MBP-NESs to CRM1 measured by differential bleaching, monitored by a microscale thermophoresis instrument. MBP-NESs compete with FITC-PKI[NES] for CRM1 in competition titrations. Fitted binding curves are overlaid onto data points with error bars representing the mean and standard deviation of triplicate titrations. Dissociation constants ($K_{DS}$) of the NESs are reported below the graphs with ranges in brackets representing the 68.3% confidence intervals. Binding of MBP-NPMmutA[NES] (a moderate CRM1 binder) and MBP-SNUPN[NES] (a weak binder) is shown on the rightmost panel for reference. *Experiments performed on separate days were fitted with a new triplicate set of direct bind titrations. (**C**) Leptomycin B (LMB) sensitive nuclear export activity of EYFP-NLS-NES fusions in HeLa cells. YFP (pseudocolored in yellow), Hoechst (pseudocolored in blue), and merged images were captured using spinning disk confocal microscope (60×). Images are maximum intensity projection of five confocal Z stacks spaced 0.3 μm apart.

The following figure supplements are available for figure 4:

**Figure supplement 1**. Binding affinities of shorter hRio2[NES] and CPEB4[NES] constructs.

**Figure supplement 2**. Differential bleaching of fluorescence probe in a sigmoidal and binding-dependent manner.

as calculated using F-statistics and error-surface projection method (*Bevington and Robinson, 1992*). When NES sequences are reversed, MBP-hRio2[NES]-R and MBP-CPEB4[NES]-R still bind CRM1 with similar affinities, $K_D$s of 2400 nM [2100,2800] and 780 nM [610,980], respectively (*Figure 4A,B*). All of the NES peptides were also cloned into EYFP-NLS-NES fusions and tested for nuclear export activity in HeLa cells. They were all found to direct nuclear export in a Leptomycin B sensitive manner, suggesting that the reverse peptides function as active NESs in cells (*Figure 4C*).

Crystal structures of CRM1-bound hRio2[NES]-R and CPEB4[NES]-R peptides were solved at 2.3 Å and 2.9 Å resolution, respectively (*Figure 5*, *Figure 5—figure supplement 1* and *Table 1*). These structures show the peptides binding in the plus direction, that is, opposite that of their wild-type

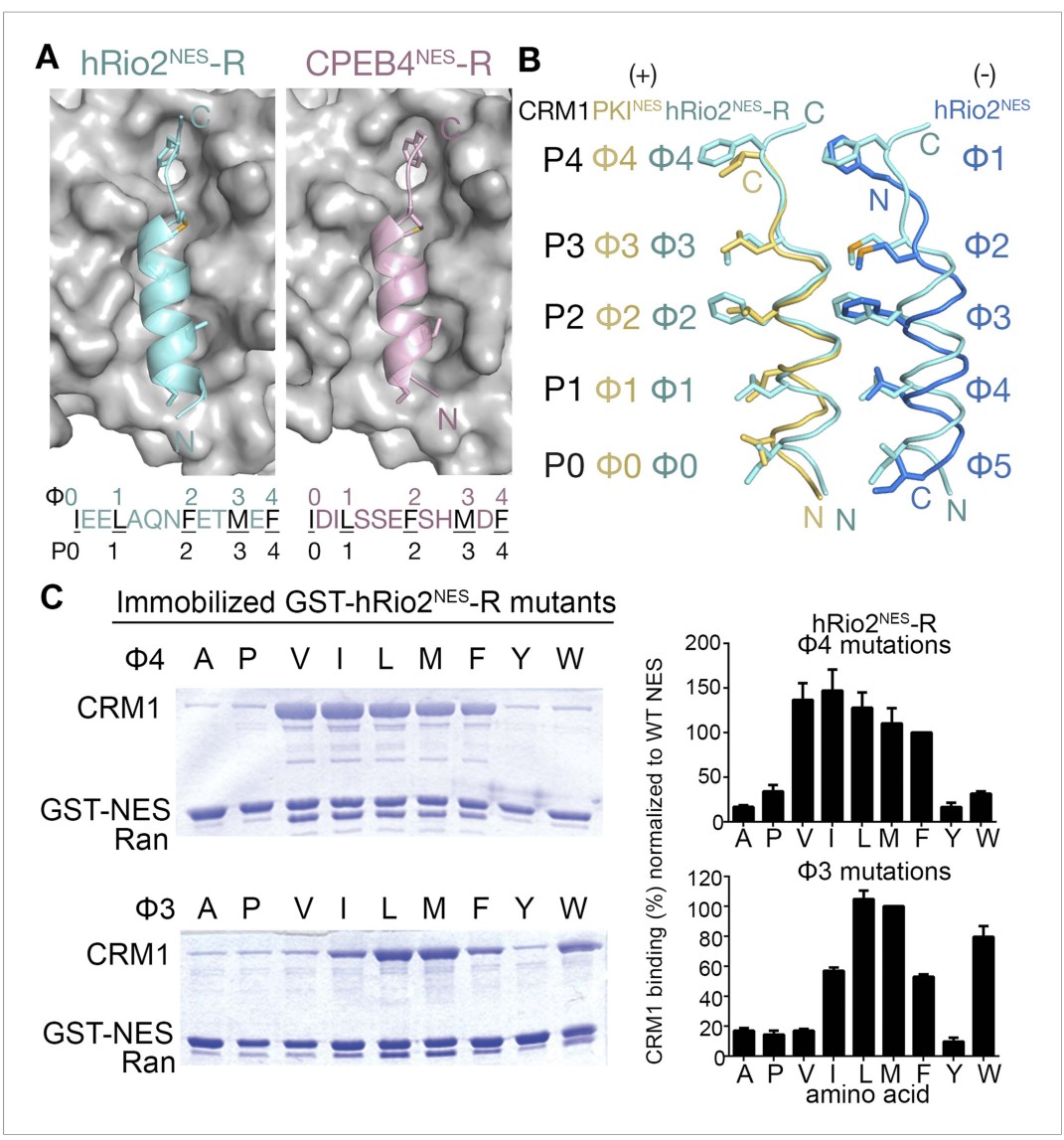

**Figure 5**. hRio2[NES]-R and CPEB4[NES]-R are plus NESs. (**A**) Structures of hRio2[NES]-R (light blue) and CPEB4[NES]-R (light pink) bound to CRM1 (gray surfaces). (**B**) Pairwise comparisons of PKI[NES] (yellow), hRio2[NES]-R (light blue), and hRio2[NES] (blue) when bound to CRM1. (**C**) In vitro pull-down assay of purified human CRM1 binding to immobilized GST-hRio2[NES]-R mutants (Φ3 or Φ4 mutated) in the presence of excess [Sc]RanGTP. Relative band intensities of triplicate experiments are plotted in histograms.

The following figure supplement is available for figure 5:

**Figure supplement 1**. Electron densities of the reverse NES peptides.

counterparts (*Figure 5A*). The CRM1 grooves in the hRio2[NES]-R and CPEB4[NES]-R complexes are almost identical to those in the wild-type hRio2[NES] and CPEB4[NES] complexes (Cα rmsds 0.2–0.3 Å). The N-terminal helices of hRio2[NES]-R and CPEB4[NES]-R that span Φ0–Φ3 bind CRM1 much like the helix of the plus direction PKI[NES]. Their C-terminal strand-like ΦXΦ segments bind in the narrow part of the CRM1 groove, but are placed slightly outward toward solvent, perhaps to better accommodate the large Phe and Met side chains in the P3 and P4 CRM1 pockets (*Figure 5B*). Pull-down assays with single amino acid mutants of hRio2[NES]-R reveal that smaller hydrophobic residues such as leucine in the Φ3 position and isoleucine, leucine, and valine in the Φ4 position are preferred for binding to CRM1 (*Figure 5C*). The preference for smaller hydrophobic side chains in ΦXΦ segment can possibly be explained by the relief of steric constraints caused by the bulky phenylalanine in the native hRio2[NES]-R sequence. The structures of CRM1-bound hRio2[NES]-R and CPEB4[NES]-R peptides support the idea that the spacing between the hydrophobic residues is critical for determining the orientation the NES binds. When the sequence and the hydrophobic spacing pattern of an NES are reversed, the direction of the peptide binding CRM1 is also reversed. However, binding affinities of the NESs are similar regardless of binding orientation, consistent with the observation that hydrophobic interactions between the CRM1 groove and side chains in the Φ positions of hRio2[NES] and CPEB4[NES], which likely govern CRM1-NES affinity, are preserved in hRio2[NES]-R and CPEB4[NES]-R.

To more rigorously test the idea that position of the ΦXΦ motif is critical for determining NES orientation, we flipped the C-terminal ΦXΦ (LDI) of PKI[NES] (SNELALKLAGLDI) to the N-terminus of the peptide while preserving the sequence of the NES helix of wild-type PKI[NES] (*Figure 6A*). We named the new peptides PKI[NES]-Flip and three variations were designed. PKI[NES]-Flip1 has the inverted wild-type LDI at the N-terminus, giving sequence **IDL**NELALKLAGL. The two hydrophobic side chains in the N-terminal Φ-X-Φ were incrementally made larger to generate PKI[NES]-Flip2 (**FDL**NELALKLAGL)

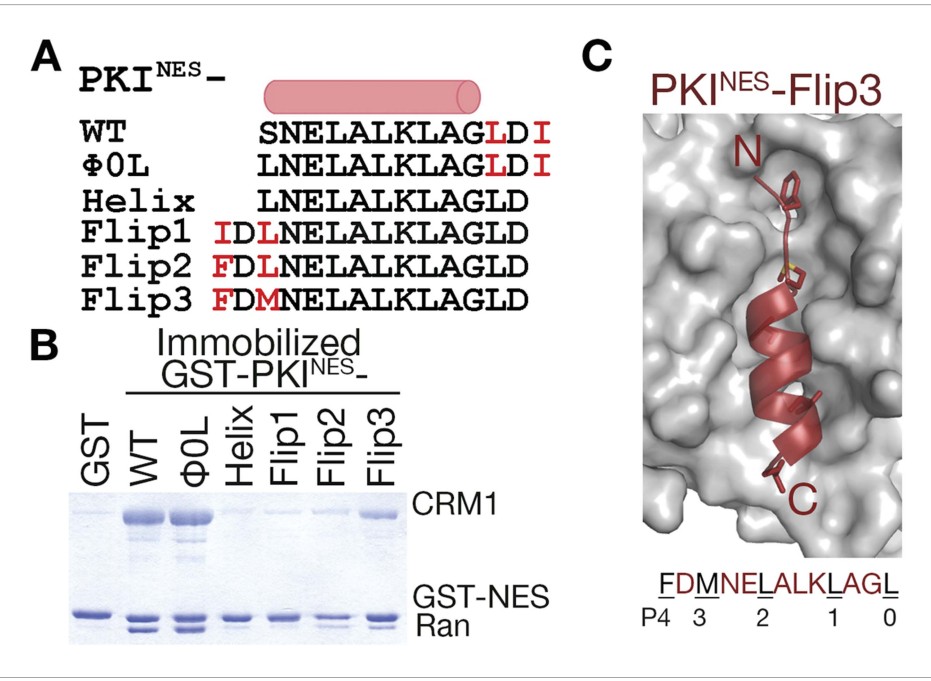

**Figure 6.** N-terminal ΦXΦ motif generates a minus orientation PKI[NES]-Flip mutant. (**A**) Sequence alignment of PKI[NES] and PKI[NES]-Flip peptides with their hydrophobic residues of ΦXΦ motifs in red and the NES helix shown as a cylinder. (**B**) Pull-down assay of immobilized GST-PKI[NES] mutants, purified CRM1 and RanGTP (Coomassie-stained SDS/PAGE). (**C**) Structure of the PKI[NES]-Flip3 peptide (red, in cartoon with Φ residues in sticks) bound to CRM1 (gray surface) with its sequence and CRM1 pockets for each Φ residue shown below.

The following figure supplement is available for figure 6:

**Figure supplement 1**. Electron densities of the PKI[NES]-Flip3 NES peptide.

and PKI[NES]-Flip3 (**FDM**NELALKLAGL) mutants. PKI[NES]-Flip1 does not interact with CRM1 in pull-down assays, while PKI[NES]-Flip2 and PKI[NES]-Flip3 show graded increases in CRM1 binding (*Figure 6B*). We solved the structure of PKI[NES]-Flip3 bound to CRM1 at 2.5 Å resolution and it indeed binds in the minus direction (*Figure 6C*, *Figure 6—figure supplement 1* and *Table 1*). These results show that NES binding in the minus vs plus direction is determined by placement of the ΦXΦ pattern at the N- or C-terminal end of the NES peptide. Secondary to this positioning, hydrophobic side chains of the N-terminal ΦXΦ segment of a minus NES should be long enough to reach into binding pockets and pack with the CRM1 groove favorably.

## Bioinformatics analysis of minus NESs in an NES database

The discovery that CRM1 binds NESs in both the plus and minus directions almost doubles the number of possible NES consensus sequences. Of the six NES patterns in *Figure 1A*, class 1a, 1b, 1c, and 1d patterns are asymmetric, whereas class 2 and class 3 patterns are symmetric. Each of the asymmetric class 1a, 1b, 1c, and 1d patterns, which represent plus NESs, could be reversed to give class 1a-R, 1b-R, 1c-R, and 1d-R patterns that represent minus NESs (*Figure 7A*). In principle, symmetric class 2 and 3 patterns can also bind CRM1 in both the plus and minus directions. For example, the class 2 Rev[NES] binds CRM1 in the plus direction as an entirely extended chain, but it is also possible that hydrophobic side chains of another class 2 NES can be presented from a similar extended peptide in the minus direction. However, it remains to be determined whether any of the currently known class 2 and true class 3 peptides can indeed bind in the minus direction. Expansion of

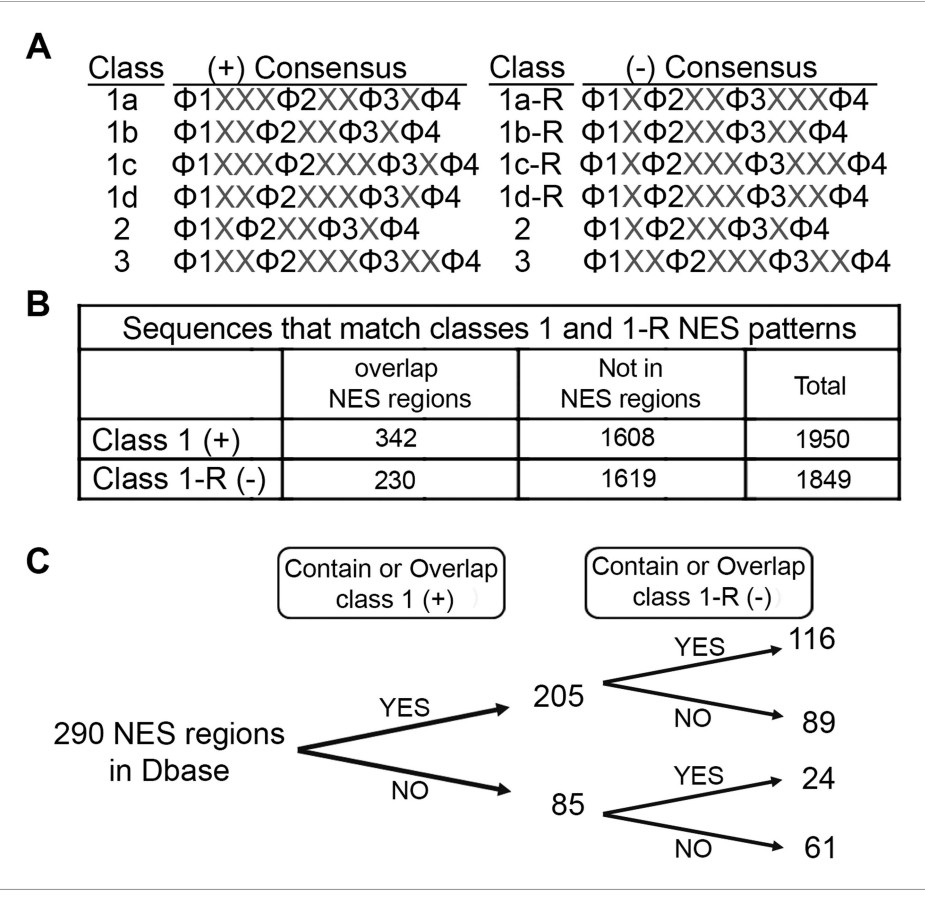

**Figure 7**. Prevalence of putative minus NESs in the Dbase data set. (**A**) Consensus patterns for minus NESs in new NES classes 1a-R to 1d-R (reverse of class 1 patterns). (**B**) The number of sequences in the 246 proteins in Dbase that match the class 1 (+) and class 1-R (−) consensus patterns. NES regions are defined according to original literature that experimentally identified CRM1 cargos and their NES regions. (**C**) The numbers of NES regions in Dbase divided into four categories according to the consensus matches they overlap with.

NES consensus by reversing class 1 NES consensus patterns to generate class 1-R patterns further suggests a corresponding increase of potential NESs in the proteome.

We searched for sequences that match class 1-R (minus) patterns in the Dbase data set, which compiled 246 NES-containing CRM1 cargos from previously published literature (*Xu et al., 2015*) . Each CRM1 cargo contains multiple sequences that match NES consensus patterns but most of these sequences are not functional export signals. Dbase reports a total of 290 experimentally identified NES regions for the 246 CRM1 cargos in the database. Matches for both class 1 (plus) and class 1-R (minus) patterns appear to be similarly prevalent in the 246 CRM1 cargos (1849 minus vs 1950 plus matches) (*Figure 7B*). However, plus patterns seem to be somewhat enriched within NES regions (340 plus vs 230 minus matches; Chi-square test, p-value = $1.378e^{-05}$) (*Figure 7B*). The bias for plus patterns in these previously reported NES regions may be a consequence of NES searches that were guided solely by the plus consensus patterns, since the minus patterns were unknown. The Dbase data set is further complicated by a lack of validation of direct CRM1-NES interactions. Only 60% of previously reported class 1 NESs that were tested recently were found to actually bind CRM1 (*Xu et al., 2012a*). Of the 290 NES regions in Dbase, 40% (116) contain sequences that match both class 1 (plus) and class 1-R (minus) patterns (*Figure 7C*). 89 NES regions match class 1 pattern exclusively and 24 match class 1-R patterns exclusively (*Figure 7C*), suggesting that there are still a significant population of putative minus NES even though the current NES annotation is biased and imperfect.

We further investigated the 24 NES regions that contained only class 1-R matches, filtering out four because of overlap with the class 2 consensus, which was previously not considered in the analysis. The remaining 20 NES regions contain 22 sequences that match class 1-R patterns, which were tested for CRM1 binding in pull-down assays. Of the 22 sequences tested, one degraded and another aggregated during purification resulting in only 20 relevant NES sequences. 10 out of the 20 putative minus NESs, or 50%, bind CRM1 (*Figure 8*). This percentage is similar to the proportion of tested class

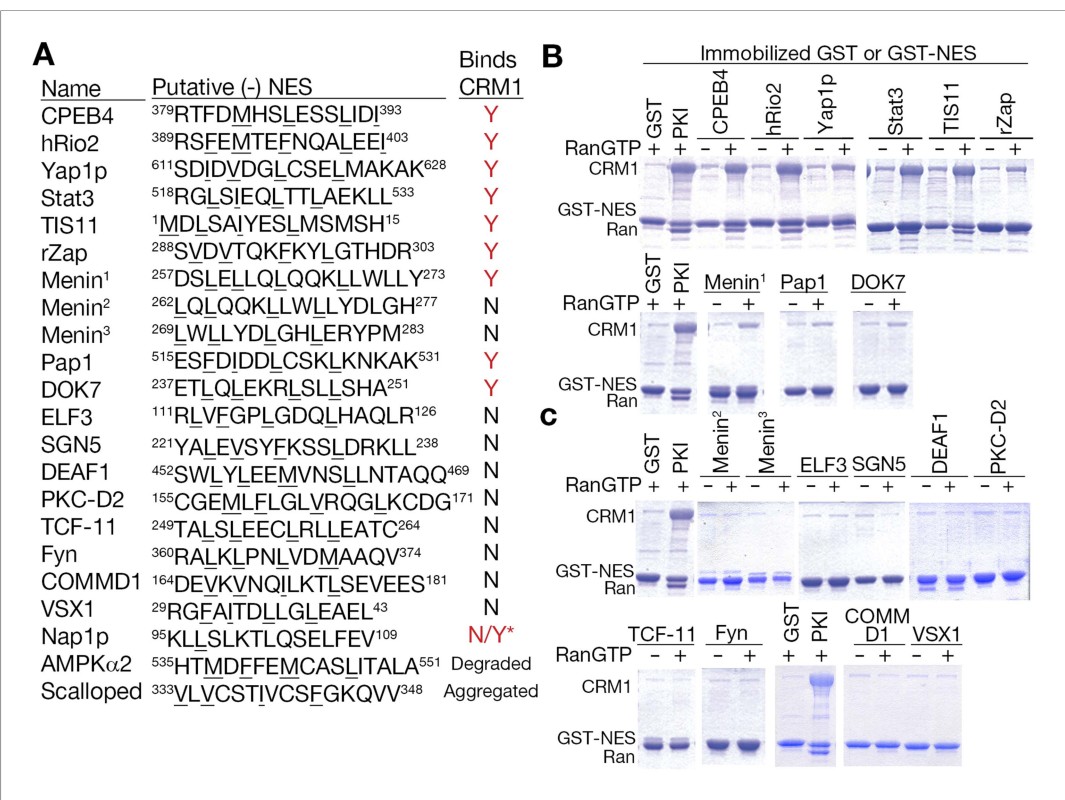

**Figure 8**. Putative minus NESs in the Dbase data set. (**A**) Summary of putative minus NESs (in the Dbase data set that match class 1-R patterns exclusively) tested for CRM1 binding. Nap1p (*) was previously shown to direct nuclear export in cells even though no CRM1 binding was observed (*Xu et al., 2015*). (**B**) Putative minus NESs that bind CRM1 in pull-down assays with CRM1 and RanGTP. (**C**) Putative minus NESs that show no observable CRM1 binding. DOI: 10.7554/eLife.10034.016

1 (plus) NESs that bind CRM1 (*Xu et al., 2012a*). These results suggest that there are a substantial number of functional NESs that likely bind CRM1 in the minus direction, even in an NES dataset like Dbase where many NESs were previously identified using the only plus NES consensus patterns. These possibly include cases where NES patterns have been mistakenly annotated or annotated as previously non-canonical patterns. This new expansion of NES consensus provides a means to identify previously unrecognizable NESs in previously identified and new CRM1 cargos.

## Discussion

The NES appears to be the only nuclear-targeting signal, and perhaps the only organelle-targeting signal, that has been shown thus far to bind its receptor in both polypeptide directions. This is in contrast to several modular-domain signaling systems, which are known to bind their linear motifs in both polypeptide chain directions (*Feng et al., 1994*; *Lim et al., 1994*; *Osawa et al., 1999*; *Swanson et al., 2004*; *Song et al., 2005*; *Lorenz et al., 2008*; *Ng et al., 2008*; *Neufeld et al., 2009*). Are protein systems that recognize linear motifs in opposite orientations unique or will most linear motifs bind their receptors in both orientations even though this phenomenon has not yet been observed for many? Alternatively, do linear motifs that bind only in a single orientation do so because of spatial constraints inherent to their cellular functions?

Regularly spaced hydrophobic pockets in the CRM1-NES groove interact with similarly spaced NES side chains that often project from one face of amphipathic α-helices. Interestingly, several other linear motifs that bind in opposite orientations (Paxillin LD motifs, HBP1/Mad1 Sin2-interaction domains, SH3-binding polyproline peptides, and various calmodulin targets [*Osawa et al., 1999*; *Swanson et al., 2004*; *Lorenz et al., 2008*; *Neufeld et al., 2009*]) also present side chains from helices for recognition. Side chain to side chain distances within secondary structural elements of motifs are preserved regardless of polypeptide orientation, thus producing a feature that may be conducive for binding in opposite orientations. The required shift of the backbone to put the plus and minus side chains in the same position is more likely for linear motifs than for extensive interfaces between two folded proteins, since the latter are constrained by many additional contacts outside of the helix-binding groove. Thus, bidirectional recognition is probably more prevalent in recognition of linear helical motifs than in recognition of larger structured elements.

Extended linear motifs such as phosphotyrosine peptides that bind SH2 domains also bind in opposite orientations (*Ng et al., 2008*). Here, side chains, mainly the phosphotyrosine side chain, contribute the majority of binding energy, which can still be preserved when peptide orientation is reversed. Linear motifs that use mostly side chains for binding may be amenable to interactions in opposite orientations but those that make extensive contacts using their backbones may be limited to a particular orientation. For example, the IBB region of Importin-α, which is the nuclear localization signal or NLS that binds directly to Importin-β, is a long 28-residue helix that is preceded by a loop (*Cingolani et al., 1999*). The IBB uses mostly charged and polar side chains to interact with Importin-β, and perhaps these side chain interactions could be preserved when polypeptide direction of the NLS peptide is flipped. Similarly, PY-NLS binding to Karyopherin-β2 (also known as Transportin-1) involves mostly the NLS side chains, and we may observe these NLSs binding to Karyopherin-β2 in the opposite orientation in the future (*Lee et al., 2006*; *Soniat et al., 2013*). In contrast, the classical-NLS recognition by Importin-α and the Kap121-specific lysine-rich NLS (also called the IK-NLS) recognition by Kap121 involve extensive interactions with the NLS main chains and are therefore less likely to bind bidirectionally to their importins (*Conti et al., 1998*; *Fontes et al., 2000*; *Kobayashi et al., 2015*; *Soniat and Chook, 2015*). Further studies will inform on orientation requirements for NLSs binding to their respective importins. We suggest that bidirectional recognition may, in fact, be widely present, but simply not widely observed.

Components of modular-domain signaling and nuclear-targeting systems consist of mostly soluble proteins that bind linear motifs found within intrinsically disordered regions. These protein–peptide interaction systems are relatively free of spatial constraints compared to systems that bind organelle-targeting signals for delivery into membrane compartments. An example of the latter is the binding of ER signal sequences by the signal recognition particle SRP, which is likely constrained spatially by the nascent chain emerging from the ribosome and by subsequent delivery into the lumen of the translocon (*Janda et al., 2010*; *Akopian et al., 2013*). In principle, linear motifs that could bind in both orientations are sometimes constrained by other factors that limit them to only one. The CRM1-NES interaction is free from such spatial constraints as the entire exportin–cargo complex enters the

nuclear pore complex for transport to the cytoplasm, thereby allowing some NESs to bind CRM1 in the plus orientation and others in the minus orientation.

Finally, accurate prediction of NESs has been difficult because of the breadth and simultaneously, the insufficient coverage of the NES consensus. Many functional NES-containing regions of proteins contain multiple NES consensus matches and sometimes no NES match, suggesting that the set of NES consensus does not provide sufficient coverage for NES identification. Our study shows that structures of NESs bound to CRM1 can accurately define consensus patterns and sometimes identify new consensus patterns. Expansion of the NES consensus upon discovery of minus NESs leads to improved coverage of potential NESs, thus allowing identification of previously unrecognized NESs in known and new CRM1 cargos. However, the improved coverage afforded by the knowledge of bidirectional NES-binding is largely orthogonal to the problems in NES prediction that arise from false positive NESs (*Fu et al., 2011*; *Xu et al., 2012a*). The majority of the NES patterns describe the ubiquitous 2-turn amphipathic helix, which are found in most helix-containing proteins, and many of these consensus-matching sequences are part of hydrophobic cores that are not accessible for CRM1 binding. In the development of NES predictors (NESsential by the Horton Lab and LocNES by the Chook lab, *Fu et al., 2011*; *Xu et al., 2015*), we found that prediction accuracy was improved by using both sequence and structural/biophysical properties (such as disorder propensity and/or solvent accessibility) as features for machine-learning methods. The latter features allowed consensus-matching sequences in the interior of folded domains to be flagged. A set of consensus sequences with high coverage rate such as the expanded set in *Figure 7A* is desirable when employed as a pre-filter in NES prediction as the machine-learning process that follows serves to reduce false positive matches. Future identification of minus NESs will also increase the size, diversity, and accuracy of experimental NES databases, which are the training/testing data sets for the development of our next generation NES predictors.

In summary, we have found that NES peptides can bind the narrow CRM1-NES groove in two opposite orientations, which we now describe as the plus and minus orientations. Whether an NES binds CRM1 in the plus or minus orientation is determined by the location of its ΦXΦ strand motif. A C-terminal ΦXΦ motif that follows a helix dictates a plus NES, while an N-terminal ΦXΦ followed by a helix results in a minus NES. The five hydrophobic pockets in the CRM1-NES groove interact with hydrophobic side chains that are presented in many different ways on NES peptides, by different secondary structural elements and in both polypeptide chain directions, to enable specific recognition of diverse NES sequences.

## Materials and methods

### Protein expression, purification, and complex formation

$^{Sc}$CRM1 (1–1058, Δ377–413, $^{537}$DLTVK$^{541}$ to GLCEQ, V441D) was cloned into the previously described pGEX-TEV vector (*Chook and Blobel, 1999*). As previously described in *Sun et al. (2013)* polypeptide segments that make up the $^{Sc}$CRM1 and human CRM1-NES grooves are 81% identical in sequence, with complete conservation in residues lining the groove that contact NESs and inhibitors (*Sun et al., 2013*). In order to maximize similarity to the human CRM1-NES groove, we mutated the only stretch of $^{Sc}$CRM1 groove residues that has more than 2 non-conserved residues, $^{537}$DLTVK$^{541}$ in the NES-binding groove of $^{Sc}$CRM1 to the human CRM1 sequence GLCEQ (*Sun et al., 2013*). Yrb1p (residues 62–201; or RanBP1) was cloned into pGEX-TEV and human Ran (full-length) was cloned into the pET-15b vector. Various NESs were cloned into the pMal-TEV vector. Sequences of NES peptides used for crystallization after TEV cleavage are hRio2$^{NES}$: GGSY$^{389}$RSFEMTEFNQALEEI$^{403}$; hRio2$^{NES}$-R: GGSYGKIEELAQNFETMEFSR; CPEB4$^{NES}$: GGSY$^{379}$RTFDMHSLESSLIDI$^{393}$: CPEB4$^{NES}$-R: GGSYRMIDILSSELSHMDFTR; PKI$^{NES}$-Flip3: GGSYRSFDMNELALKLAGLD. Sequences of NESs used for binding affinity measurements are listed in *Figure 4*. All proteins were expressed separately in *E. coli* BL-21(DE3) by induction with 0.5 mM isopropyl β-D-1-thiogalactopyranoside for 10 hr at 25°C. GST-$^{Sc}$CRM1 and GST-RanBP1 cells were lyzed in buffer containing 40 mM HEPES (pH 7.5), 2 mM MgOAc, 200 mM NaCl, 10 mM dithiothreitol (DTT) and protease inhibitors, purified by affinity chromatography using glutathione Sepharose 4B beads (GE Healthcare Life Sciences, PA), followed by cleavage with TEV protease and finally size-exclusion chromatography in GF buffer (20 mM HEPES pH 7.5, 100 mM NaCl, 5 mM MgOAc, and 2 mM DTT). Cells expressing His-Ran were lyzed in buffer containing 50 mM HEPES (pH 8.0), 2 mM MgOAc, 200 mM NaCl, 10% (vol/vol) glycerol, 5 mM imidazole (pH 7.8), 2 mM DTT and protease inhibitors, purified by

affinity chromatography with Ni-NTA Agarose (Qiagen, Hilden, Germany) and further purified by gel filtration chromatography in TB buffer (20 mM HEPES pH 7.5, 110 mM KOAc, 2 mM MgOAc, 10% glycerol, and 2 mM DTT). Ran was loaded with non-hydrolyzable GTP analog GppNHp by nucleotide exchange. Cells expressing MBP-NESs were lyzed in buffer containing 50 mM HEPES pH 7.5, 100 mM NaCl, 10% glycerol, 2 mM DTT and protease inhibitors, purified by affinity chromatography using amylose resin (New England Biolabs, MA) and ion exchange chromatography using (HiTrap Q, GE Healthcare Life Sciences) with a salt gradient from 50 mM to 1 M NaCl. Purified MBP-NES proteins were concentrated, cleaved with TEV protease and NES peptides were then isolated by gel filtration chromatography in GF buffer. To assemble the CRM1-Ran-RanBP1-NES complex, the RanGppNHp-RanBP1 heterodimer was first purified by gel filtration chromatography. $^{Sc}$CRM1*, Ran-RanBP1 and NES peptides were then assembled in 1:3:10 molar ratio and the quaternery complexes were purified by gel filtration chromatography in GF buffer. Purified $^{Sc}$CRM1*-Ran-RanBP1-NES complexes were concentrated to ~10 mg/ml and excess NES peptides were added to stabilize the complex during concentration.

## Crystallization, data collection, and structure determination

$^{Sc}$CRM1-Ran-RanBP1-NES complexes were crystallized in 17% (wt/vol) PEG3350, 100 mM Bis-Tris (pH 6.4), 200 mM ammonium nitrate, and 10 mM Spermine HCl. Crystals were cryoprotected with the same crystallization condition supplemented with up to 23% PEG3350 and 12% glycerol and flash cooled in liquid nitrogen. X-ray diffraction data were collected at 0.9795 Å at the Advanced Photon Source 19ID beamline in the Structural Biology Center at Argonne National Laboratory. Data were indexed, integrated, and scaled using HKL-3000 (*Minor et al., 2006*). All crystals in this study were isomorphous to crystals of previously solved inhibitor-bound and unliganded $^{Sc}$CRM1-Ran-RanBP1 complexes and has space group P4$_3$2$_1$2. Therefore, structures were determined by multiple rounds of refinement of unliganded complex (4HB2) against collected data using PHENIX (*Adams et al., 2010*; *Afonine et al., 2012*) and manual modeling in Coot (*Emsley et al., 2010*). X-ray/stereochemistry and X-ray/ADP weights were optimized in phenix.refine in final stages of refinement. Structure validation was guided by Molprobity suite in PHENIX (*Chen et al., 2010*). Ramachandran plots of the five structures showed that 97.3–97.9% of residues are in favored regions and 0.0–0.1% are in disallowed regions. Structure figures were generated with PyMOL (*Schrodinger, 2010*). NESs in *Figures 2 and 5* were compared by superimposing H12A helices of their respective CRM1s.

## In vitro CRM1-NES pull-down binding assays

Full-length human CRM1 ($^{Hs}$CRM1) was purified in the same manner as $^{Sc}$CRM1* with buffers supplemented with 10% glycerol. $^{Sc}$Ran (Gsp1p) was expressed using pET21d-GSP1 (GSP1 residues 1–179, Q71L) (gift from Dr. Takuya Yoshizawa) and purified as described above for human Ran (buffers in HEPES pH 7.4 instead of pH 8.0). After affinity purification, $^{Sc}$Ran was loaded with GTP (incubated with molar excess of ethylenediaminetetraacetic acid (EDTA) for 30 min on ice followed by incubation with excess GTP and MgOAC for 30 min at room temperature) and then purified by ion exchange chromatography (HiTrap SP, GE Healthcare Life Sciences). NESs were cloned into the pGEX-TEV vector (*Chook and Blobel, 1999*), purified, and immobilized on glutathione Sepharose beads (GE Healthcare Life Sciences) in TB buffer described above containing 15% glycerol. 2.5 µM $^{Hs}$CRM1 and 7.5 µM $^{Sc}$RanGTP were added to ~10 µg of immobilized GST-NESs in TB buffer in total volumes of 200 µl for 30 min at 4°C. Unbound proteins were washed extensively with TB buffer and bound proteins on the Sepharose beads were separated by sodium dodecyl sulfate polyacrylamide gel electrophoresis (SDS/PAGE) and visualized with Coomassie Blue staining. All binding assays were performed in triplicates. To compare the relative intensities of CRM1 bands to yield an estimate of binding activities of various NES mutants, SDS/PAGE gels were dried and scanned with an Epson V300 scanner and the images analyzed with ImageJ software. CRM1 band intensities were corrected for differences in GST-NESs band intensities and normalized to wild-type control intensity in each set of mutations. Corrected relative CRM1 band intensities were plotted as histograms with standard errors with GraphPad Prism.

To test putative minus NESs identified from the Dbase data set, 5 µM $^{Hs}$CRM1 with or without 15 µM $^{Sc}$RanGTP were used instead. Putative NESs that show no CRM1 binding were expressed in larger scale and purified by size-exclusion chromatography to assess their aggregation states. They were also subjected to intact mass determination by mass spectroscopy to ensure that the GST-NES proteins were not degraded.

## Nuclear-cytoplasmic localization assays

Cellular localization of EYFP2-NLS-NES fusion proteins overexpressed in HeLa cells was observed using procedures as previously described (*Xu et al., 2015*). Expression constructs for EYFP2-NLS-NES fusion proteins were cloned similarly into pEYFP2-SV40$^{NLS}$ vectors. Live cell images were collected using a spinning disk confocal microscope system, Nikon-Andor (Nikon, NY), and MetaMorph software. Image analysis was performed similarly with ImageJ. CRM1 dependence was demonstrated by the nuclear accumulation of EYFP fusion proteins after treatment with 2 nM Leptomycin B for 16 hr at 37°C. Experiments were performed in at least duplicates with over a total of 150 transfected cells.

## Competition differential bleaching assay monitored by microscale thermophoresis to measure CRM1-NES affinities

Differential bleaching was used as a parameter to monitor binding of MBP-NES proteins to CRM1. In short, we observed that the fluorophore reporter attached to PKI$^{NES}$ (FITC) underwent a reproducible time-dependent bleaching when exposed to excitation light (*Figure 4—figure supplement 2*). Furthermore, this phenomenon was concentration-dependent, that is, the bleaching was accelerated when the FITC-PKI$^{NES}$ probe was exposed to increasing concentrations of CRM1. However, this phenomenon was saturable at high concentrations of CRM1, indicating that it was a function of CRM1 binding and not simply the presence of the protein that was causing the change. The differential bleaching can be counteracted by titrating of mixture of FITC-PKI$^{NES}$ and CRM1 with a known competitor, MBP-PKI$^{NES}$, which competes directly with the fluorescent probe for the NES binding groove in CRM1. A sigmoidal appearance of the binding and competition isotherms is observed when differential bleaching, quantified as the average fluorescence at a time after bleaching normalized by the averaged fluorescence just after the beginning of bleaching, is plotted vs titrant concentration in a semilog graph (see *Figure 4*).This illustrates that this bleaching behavior can be described as a two-state system where unbound probe and CRM1-bound probe bleach at different but specific rates, and that these quantities report on the populations of bound and unbound FITC-PKI$^{NES}$. A detailed description of the data-fitting procedures will be described in manuscript in preparation by C.A.B. For error reporting, we used F-statistics and error-surface projection method to calculate the 68.3% confidence intervals of the fitted data (*Bevington and Robinson, 1992*). While error reporting using the error surface projection method is relatively uncommon, the ranges more accurately represent the true confidence intervals given the observed noise in the performed set of experiments because they explicitly account for the ability of the fitting algorithm to compensate for fitting defects by modifying correlated parameters. Thus, they provide better evaluation of the fitted data than other, more commonly used methods (e.g., error estimations from the parametric variance-covariance matrix).

All proteins used were subjected to an extra gel filtration step and dialysis overnight in TB buffer with 15% glycerol to remove possible aggregation and ensures buffer matching. The FITC-PKI$^{NES}$ peptide (FITC-SGNSNELALKLAGLDINKT) was chemically synthesized by GenScript, NJ and dissolved in the TB buffer with 15% glycerol. For the direct titration, $^{Hs}$CRM1 was serially diluted from 40 µM to 1.2 nM and incubated with mixture of 120 µM $^{Sc}$Ran-GTP and 40 nM FITC-PKI$^{NES}$ in 1:1 vol to a total volume of 20 µl, and incubated for 1 hr in the dark at room temperature. For competition experiments, MBP-NESs were serially diluted from 100 µM to 3 nM in presence of 40 nM of FITC-PKI$^{NES}$ and incubated with mixture of 300 nM $^{Hs}$CRM1 and 120 µM $^{Sc}$Ran-GTP in 1:1 vol to a total of 20 µl, and incubated for 1.5 hr in the dark at room temperature. All reactions mixtures were supplemented with 0.05% Tween-20. Following incubations, reactions were loaded into NanoTemper's 'Standard' treated capillaries and fluorescence signals were monitored by NanoTemper Monolith NT.115 equipment (NanoTemper Technologies, München, Germany) with 60% LED power for 10 s. Titrations for parallel comparisons were performed in triplicates on the same day. Data collected were then analyzed with PALMIST (manuscript in preparation) and imported to GUSSI for generating figures (*Brautigam, 2015*).

## Bioinformatic studies of plus and minus NESs in the Dbase data set

Protein sequences in the Dbase data set, a non-redundant compilation of CRM1 cargos from two of the most recent NES databases, ValidNESs (*Fu et al., 2013*), and NESdb (*Xu, Grishin et al., 2012b*) (http://prodata.swmed.edu/LRNes), were used for analyses of plus and minus NESs. All protein sequences in the Dbase data set were first compiled along with their annotated NES regions into an in-house database implemented by MySQL (version 5.5.43) on Linux (Ubuntu 12.04). NES regions are

defined according to original reports in the published literature that identified the CRM1 cargos. PHP (version 5.3) regular expression with look-ahead assertions was used to capture all sequences (including overlapping sequences) that match the eight different class 1 (plus) and class 1-R (minus) NES consensus patterns. Duplicate matches (such as 10-mer sequences that simultaneously match both class 1a and 1d patterns, or match both 1a-R and 1d-R patterns) were removed using Linux command line tools, and the resulting numbers of consensus matches (see *Figure 7*) were used for the enrichment test of plus consensus patterns within NES regions by Chi-square test using R (version 2.14.1). The same MySQL database was used to search for putative minus NESs, and 24 NES regions that match the 1-R patterns exclusively were identified. Four of these NES regions were removed because of overlap with class 2 patterns, resulting in 20 NES regions (containing 22 1-R consensus matches) for examination of CRM1 interactions by pull-down binding assays.

## Accession codes

Structures and crystallographic data have been deposited at the PDB: 5DHF (CRM1-hRio2$^{NES}$ complex), 5DIF (CRM1-CPEB4$^{NES}$ complex), 5DI9 (CRM1-hRio2$^{NES}$-R complex), 5DHA (CRM1-CPEB4$^{NES}$-R complex), 5DH9 (CRM1-PKI$^{NES}$-Flip3 complex).

## Acknowledgements

We thank members of the Structural Biology Laboratory and Macromolecular Biophysics Resource at UTSW for crystallographic and biochemical data collection assistance, M Soniat, M Rosen, and N Grishin for comments. The use of SBC 19ID beamline at Advanced Photon Source is supported by U.S. Department of Energy contract DE-AC02-06CH11357. This work is funded by Cancer Prevention Research Institute of Texas (CPRIT) Grants RP120352 and RP150053 (YMC), R01 GM069909 (YMC), the University of Texas Southwestern Endowed Scholars Program (YMC), Welch Foundation Grant I-1532 (YMC), Leukemia and Lymphoma Society Scholar Award (YMC), and a Croucher Foundation Scholarship (HYJF).

# Additional information

## Funding

| Funder | Grant reference | Author |
| --- | --- | --- |
| Cancer Prevention and Research Institute of Texas (CPRIT) | RP120352, RP150053 | Yuh Min Chook |
| National Institutes of Health (NIH) | GM069909 | Yuh Min Chook |
| University of Texas Southwestern Medical Center (UT Southwestern) | Endowed Scholars Program | Yuh Min Chook |
| Welch Foundation (Robert A. Welch Foundation) | I-1532 | Yuh Min Chook |
| Leukemia and Lymphoma Society (LLS) | Scholar Award | Yuh Min Chook |
| Croucher Foundation | Graduate Student Scholarship | Ho Yee Joyce Fung |

The funders had no role in study design, data collection and interpretation, or the decision to submit the work for publication.

## Author contributions

HYJF, S-F, Conception and design, Acquisition of data, Analysis and interpretation of data, Drafting or revising the article; CAB, Analysis and interpretation of data, Drafting or revising the article; YMC, Conception and design, Analysis and interpretation of data, Drafting or revising the article

## Author ORCIDs

Ho Yee Joyce Fung, http://orcid.org/0000-0002-0502-1957

## Additional files

### Major dataset
The following previously published dataset was used:

| Author(s) | Year | Dataset title | Dataset ID and/or URL | Database, license, and accessibility information |
|---|---|---|---|---|
| Xu D, Marquis K, Pei J, Fu SC, Cağatay T, Grishin NV, Chook YM | 2014 | LocNES: a computational tool for locating classical NESs in CRM1 cargo proteins | http://dx.doi.org/ 10.1093/bioinformatics/ btu826 | Publicly available in Supplementary Data of the paper at Bioinformatics. |

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
