## [Decision Letter]

Thank you for submitting your work entitled “Structural determinants of Nuclear Export Signal orientation in binding CRM1” for peer review at *eLife*. Your submission has been favorably evaluated by John Kuriyan (Senior Editor), Mingjie Zhang (Guest Reviewing Editor), Douglas Barrick, and Gino Cingolani (peer reviewers).

The reviewers have discussed the reviews with one another, and the Reviewing editor has drafted this decision to help you prepare a revised submission.

This manuscript describes the binding of a new class of nuclear export signals (here Rio2 and CPEB4) to exporter CRM1. The typical hydrophobic spacing in these new signals differ from the crystallographically defined mechanism of recognition of protein kinase-A inhibitor (PKI), and using x-ray crystallography, the authors find that the orientation of the new CRM1s are reversed (N to C) when bound to the pocket. In retrospect, bidirectional recognition of NES is not surprising if one considers that CRM1 does not make contacts with the NES main-chain, unlike importins (e.g. Importin-α, β, etc.) that make extensive contacts with the NLS backbone, hence forcing the NLS side chains in an orientation-dependent conformation. The authors then deduce the spacing rules for orientation, and show that they can reverse the orientation of both PKI, and the Rio2 and PKI sequences. The authors also define specificities of hydrophobic binding residues qualitatively using pull-down assays. Finally, using their modified sequence profile, they identify a new potential set of sequences and show that a significant subset of them bind. Overall, this is an interesting and fairly complete study that is important to understand nuclear export and molecular recognition. It should have broad interest, from cell biologists to biophysicists.

Essential revisions:

The authors should address the following questions and comments in the form of revision.

1) The authors should try to expand on the correlation between NES directionality of binding to CRM1 and K_D_ of NESs for CRM1. Can something be said in this regard? Can directionality of binding help explain the wide range of K_Ds_ measured for NES binding to CRM1?

2) Related to the above point, the authors need to give uncertainties on their binding constants (subsection “Structural determinants of the plus versus minus NES”, first paragraph and Figure 4). Also, the way that the authors report K_D_ values is rather uncommon. It will be easier for readers to understand if conventional curve fitting errors are reported.

3) Regarding the bioinformatics, consensus sequences (five regularly spaced hydrophobic residues) are not a very stringent search pattern. The chances that such sequences will appear in random sequence (and non-import sequences) are quite high. And maybe it should be even higher in folded domains where hydrophobic residues show this kind of periodicity to form amphipathic secondary structures. The authors should comment about these false positives, and how likely they are to influence their numbers. Would it be possible if additional criteria can be included in improve the prediction power? For example, the predicted NESs should not be a part of folded protein structure.

4) The authors are suggested to include at least one omit map in the main text, as this validation is essential to determine the directionality of NES binding to CRM1.

---

## [Author Response]

*This manuscript describes the binding of a new class of nuclear export signals (here Rio2 and CPEB4) to exporter CRM1. The typical hydrophobic spacing in these new signals differ from the crystallographically defined mechanism of recognition of protein kinase-A inhibitor (PKI), and using x-ray crystallography, the authors find that the orientation of the new CRM1s are reversed (N to C) when bound to the pocket. In retrospect, bidirectional recognition of NES is not surprising if one considers that CRM1 does not make contacts with the NES main-chain, unlike importins (e.g. Importin-α, β, etc.) that make extensive contacts with the NLS backbone, hence forcing the NLS side chains in an orientation-dependent conformation. The authors then deduce the spacing rules for orientation, and show that they can reverse the orientation of both PKI, and the Rio2 and PKI sequences. The authors also define specificities of hydrophobic binding residues qualitatively using pull-down assays. Finally, using their modified sequence profile, they identify a new potential set of sequences and show that a significant subset of them bind. Overall, this is an interesting and fairly complete study that is important to understand nuclear export and molecular recognition. It should have broad interest, from cell biologists to biophysicists*.

We thank the reviewers for their insight about importins that make extensive contacts with the NLS backbone, which may explain orientation-dependent interactions of the NLS. The comment above has inspired us to add a short discussion about interactions between various types of NLSs (classical-NLS, Kap121-specific NLS, PY-NLS and the IBB) with their importins, and how the interactions might affect NLS directionalities on the importins (please see in the Discussion section: “Extended linear motifs such as phosphotyrosine peptides […] may, in fact, be widely present, but simply not widely observed”).

*Essential revisions*:

The authors should address the following questions and comments in the form of revision.

*1) The authors should try to expand on the correlation between NES directionality of binding to CRM1 and K*_*D*_
*of NESs for CRM1. Can something be said in this regard? Can directionality of binding help explain the wide range of K*_*Ds*_
*measured for NES binding to CRM1?*

We have followed the reviewers' suggestion and further investigated the correlation of NES directionalities and their binding affinities to CRM1. In the original paper, we reported binding affinities of hRio2^NES^-R and CPEB4^NES^-R constructs that are two residues longer at the C-termini than constructs of their wild-type counterparts hRio2^NES^ and CPEB4^NES^ (see sequences below). We have corrected the inconsistency of comparing constructs of different lengths and repeated the MST experiments with longer versions of hRio2^NES^ and CPEB4^NES^ that are now the exact inverse counterparts of the hRio2^NES^-R and CPEB4^NES^-R constructs. Sequences of old and new constructs are listed below:

Construct 1 hRio2^NES^ (original manuscript): RSFEMTEFNQALEEI

Construct 2 CPEB4^NES^ (original manuscript): RTFDMHSLESSLIDI

Construct 3 hRio2^NES^ (new construct): RSFEMTEFNQALEEIKG

Construct 4 CPEB4^NES^ (new construct): RTFDMHSLESSLIDIMR

Construct 5 hRio2^NES^-R (original manuscript): GKIEELAQNFETMEFSR

Construct 6 CPEB4^NES^-R (original manuscript): RMIDILSSELSHMDFTR

New experiments comparing new constructs 3 and 4 with original constructs 5 and 6 show that affinities of NESs are similar regardless of binding orientation. These data support structural observations that hydrophobic interactions, which likely dominate CRM1-NES affinity, of the CRM1 groove with both plus and minus NESs are similar. Minus NESs do not appear to be systematically weaker binders compared to plus NES. However, the ability of NESs to bind in both directions likely allows more combinations of hydrophobic residues in the peptides that are viable for favorable interaction with CRM1 and the larger variation of hydrophobic residues in turn contributes to a wide range of K_D_s.

We have replaced old affinity data for hRio2^NES^ and CPEB4^NES^ with new data for the longer constructs in Figure 4. We have also changed the text in subsection “Structural determinants of the plus versus minus NES” to report the new comparisons: “Wild-type NESs MBP-hRio2^NES^ and MBP-CPEB4^NES^ bind CRM1 with K_Ds_ of 2200nM [1600,2900] and 590nM [400,840], respectively (Figure 4). The ranges in brackets represent the 68.3% confidence intervals as calculated using F-statistics and error-surface projection method (4). When NES sequences are reversed, MBP-hRio2^NES^-R and MBP-CPEB4^NES^-R still bind CRM1 with similar affinities, K_Ds_ of 2400nM [2100,2800] and 780nM [610,980], respectively (Figure 4).”

We have also added a sentence to the aforementioned subsection to connect comparison of affinities with structural comparisons:

“However, binding affinities of the NESs are similar regardless of binding orientation, consistent with the observation that hydrophobic interactions between the CRM1 groove and side chains in the Φ positions of hRio2^NES^ and CPEB4^NES^, which likely govern CRM1-NES affinity, are preserved in hRio2^NES^-R and CPEB4^NES^-R.”

Finally, affinity data for the old and shorter constructs of hRio2^NES^ and CPEB4^NES^ (constructs 1 and 2) are now reported in Figure 4—figure supplement 1, and a discussion about their affinities compared to the longer constructs (hRio2^NES^ constructs 1 vs. 3; and CPEB4^NES^ constructs 2 vs. 4) is present in the figure legend.

*2) Related to the above point, the authors need to give uncertainties on their binding constants (subsection “Structural determinants of the plus versus minus NES”, first paragraph and*
Figure 4*). Also, the way that the authors report K*_*D*_
*values is rather uncommon. It will be easier for readers to understand if conventional curve fitting errors are reported*.

We apologize for the inadvertent omission of an explanation of the error used in the original Figure 4. We now include a brief description and a citation for the statistical method that was used to calculate the ranges reported along with the fitted K_Ds_ in the main text, in subsection “Structural determinants of the plus versus minus NES”, that report the binding affinities of the various NESs: “Wild-type NESs MBP-hRio2^NES^ and MBP-CPEB4^NES^ bind CRM1 with K_Ds_ of 2200nM [1600,2900] and 590nM [400,840], respectively (Figure 4). The ranges in brackets represent the 68.3% confidence intervals as calculated using F-statistics and error-surface projection method (4).”

We also included a longer description/explanation in the Methods (subsection “Competition differential bleaching assay monitored by Microscale Thermophoresis to measure CRM1-NES affinities”).

*3) Regarding the bioinformatics, consensus sequences (five regularly spaced hydrophobic residues) are not a very stringent search pattern. The chances that such sequences will appear in random sequence (and non-import sequences) are quite high. And maybe it should be even higher in folded domains where hydrophobic residues show this kind of periodicity to form amphipathic secondary structures. The authors should comment about these false positives, and how likely they are to influence their numbers. Would it be possible if additional criteria can be included in improve the prediction power? For example, the predicted NESs should not be a part of folded protein structure*.

We strongly agree that using consensus sequences alone is not sufficient for NES prediction. In fact, we have discussed and addressed this issue at length in our previous papers on NES prediction ([17]; Xu et al., 2012; [49]). In those studies, we incorporated disorder prediction into our NES predictors, NESsential and LocNES, such that NES candidates are deemed less probable NESs if not located in disordered regions of proteins. We have revised the current manuscript to include a summary of our work on NES prediction (Discussion) where the use of sequence patterns along with incorporation of biophysical/structural properties such as disorder propensity and predicted solvent accessibility have significantly improved NES prediction. However, the prediction accuracy of our most recent and best NES predictor (LocNES) is still far from ideal (49). We have observed that structural information of CRM1 cargos can be useful to exclude NES candidates that reside within folded domains and thus are most likely inaccessible to CRM1. We plan to include more 3D structural information into our NES prediction schemes as more PDB entries of CRM1 cargos become available and modeling of 3D structures becomes more facile. In the meantime, we rephrased the Discussion accordingly (“Finally, accurate prediction of NESs has been difficult […] are the training/testing datasets for the development of our next generation NES predictors).

*4) The authors are suggested to include at least one omit map in the main text, as this validation is essential to determine the directionality of NES binding to CRM1*.

We have included omit maps for both hRio2^NES^ and CPEB4^NES^ in the new Figure 1.